# A mutual information based R-vine copula strategy to estimate VaR in high frequency stock market data

**Charu Sharma**[ID]®*, **Niteesh Sahni**®

Department of Mathematics, Shiv Nadar University, Uttar Pradesh, India

® These authors contributed equally to this work.
* charu.sharma@snu.edu.

**Data Availability Statement:** Raw data cannot be shared publicly because of copyright agreement with NSE Data & Analytics, formerly known DotEx International Ltd.(third party). Data are available from the NSE Data & Analytics, formerly known

## Abstract

In this paper, we explore mutual information based stock networks to build regular vine copula structure on high frequency log returns of stocks and use it for the estimation of Value at Risk (VaR) of a portfolio of stocks. Our model is a data driven model that learns from a high frequency time series data of log returns of top 50 stocks listed on the National Stock Exchange (NSE) in India for the year 2014. The Ljung-Box test revealed the presence of Autocorrelation as well as Heteroscedasticity in the underlying time series data. Analysing the goodness of fit of a number of variants of the GARCH model on each working day of the year 2014, that is, 229 days in all, it was observed that ARMA(1,1)-EGARCH(1,1) demonstrated the best fit. The joint probability distribution of the portfolio is computed by constructed an R-Vine copula structure on the data with the mutual information guided minimum spanning tree as the key building block. The joint PDF is then fed into the Monte-Carlo simulation procedure to compute the VaR. If we replace the mutual information by the Kendall's Tau in the construction of the R-Vine copula structure, the resulting VaR estimations were found to be inferior suggesting the presence of non-linear relationships among stock returns.

## 1) Introduction

Developing multivariate models and estimating joint density function is an area of key interest amongst researchers not only in finance but also in various other fields [1–3]. In finance, the researchers have already discarded multivariate Gaussian distributions on log returns of stocks and hence developing methods to estimate the joint distribution of stock returns have always attracted lot of interest [4]. In this paper we use Copula functions to achieve the important goal of estimating the joint probability distribution of the portfolio. The Sklar's theorem [5] expresses a multivariate cumulative distribution function in terms of the univariate cumulative distribution functions and a Copula function. So we need to overcome two challenges: firstly, to identify the probability distributions of the individual stocks and secondly devise a computationally efficient method of combining these marginal distributions with an appropriate

DotEx International Ltd., (contact via dotex_kraops@nse.co.in) for researchers who meet the criteria for access to confidential data. The terms of purchase prohibit us from redistributing the Historical Data or any component of it. However, the raw data can be purchased from NSE Data & Analytics at: https://www.nseindia.com/supra_global/content/dotex/data_products.htm We also confirm that we did not have any special access privileges that others would not have. However, we also confirm that all the processed data corresponding to each figure and each table is uploaded as a supporting file named "S1 Table".

**Funding:** The author(s) received no specific funding for this work.

**Competing interests:** NO authors have competing interests.

Copula to obtain the joint distribution of the portfolio. The Kolmogorov-Smirnov test [6] suggested the Student's t-distribution as a good choice for the probability distribution functions of the individual stocks. The second step was handled using the R-Vine Copula structure which was originally introduced in [7–9] by extending the concept of Markov Trees. Two special subclasses of R-vine copulas namely the D-vine and C-vine copulas were studied in [10] and since then have become immensely popular in the analysis of financial data owing to their simple structure [11–14]. Working with a general R-vine structure is computationally challenging especially in higher dimensions. The sequential algorithm due to Dißmann et.al. [15] is a breakthrough in this direction and enables an efficient construction of general R-vine copula structures in higher dimensions. In [15], joint distribution functions of 16 variables is computed and in this paper we go as high as 50 variables via this algorithm. It is relevant to point out that the construction of the R-vine structure in [15] made use of Kendall's Tau–a non parametric measure which captures an ordinal relationship between two random variables and also indicates a non linear relationship among them. In [16–18], this approach has been applied successfully to a number of financial markets. Some very recent works [19–23] reveal the growing popularity of mutual information between two random variables as a quantifier of a linear or non-linear relationship. Mutual information (MI) between two random variables is defined to be the relative entropy between the joint distribution and the product of the marginal distributions. A direct consequence of this definition is that MI of independent random variables is zero. MI captures the reduction in the uncertainty of one random variable given the knowledge about another random variable. In particular, Sharma and Habib [23], in the context of high frequency data, have demonstrated that mutual information based methods capture the non-linear relationship between log returns better when compared to Spearman correlation based methods. This observation motivates a key part of the present paper which deals with the computation of the joint density function of log returns of stocks using a mutual information based R-vine copula structure.

Our analysis begins with the removal of Autocorrelation and Hetroscedasticity using the GARCH models on the time series data of log returns. A number of popular GARCH models were fitted and the best of the lot turned out to be ARMA(1,1)-EGARCH(1,1). Next, the R-Vine copula structure was constructed using the error (residual) terms of the ARMA(1,1)-EGARCH(1,1) model. This approach is similar to the one taken in [18] in which data of daily returns of 96 stocks listed on S&P was analysed.

The above R-Vine structure is then used to estimate the Value at Risk (VaR) of portfolios through Monte-Carlo simulation. Recently, multivariate copula based models for the estimation of VaR have been proposed in [24] and a Kendall's Tau based vine copula model for estimating Var is presented in [18]. For earlier models focused towards the VaR estimation, the reader is referred to [25, 26]. However, none of these models have employed mutual information.

We have considered 5% and 10% VaRs for portfolios consisting 5, 10, 25 and 50 stocks in our analysis.

The remaining part of the paper is divided into 4 sections. In section 2, we give a brief description of the data used in our analysis. In section 3, we give an overview of the methods and methodology used. In section 4, we compare the effectiveness of VaR estimation based on Kendall's Tau method and MI method. In the last section, we summarize our observations and findings.

## 2) Data description

The high frequency data analysed in the present paper is an instant-by-instant record of the prices and volume of all the stocks listed on CNX100 index of the National Stock Exchange

(NSE) for each working day of the year 2014. The working hours of the NSE are from 9:00AM till 4:00PM. Further, we divide this duration into time intervals of length 30 seconds and call each such interval as a tick. The interval length of 30 seconds ensures sufficiently many data points for the fitted models to have small bias. We chose to ignore the first and the last half an hour (that is, 9–9:30AM and 3:30-4PM) data due to some ambiguity and incompleteness in the recorded data. Thus the total number of ticks considered for each working day will be 720. In general, corresponding to any tick $t$, that is, in the $t^{th}$ 30-second interval there will be several transactions for various stocks. Let $v_{i,k}^t$ be the volume of the stock $k$ traded at an instant $i$ (within the duration corresponding to the tick $t$) and $S_{i,k}^t$ be the price of stock $k$ at the instant $i$. We now define the volume weighted average price $S_{VWAP}(t,k)$ for the tick $t$ by

$$S_{VWAP}(t,k) = \frac{\sum_i v_{i,k}^t S_{i,k}^t}{\sum_i v_{i,k}^t}. \tag{1}$$

Here the summation runs over all possible instances within the 30-second duration corresponding to the tick $t$. The log return of each stock $k$ at tick t is given by

$$R_{t+1,k} = \ln\big(S_{VWAP}(t+1,k)\big) - \ln\big(S_{VWAP}(t,k)\big) \tag{2}$$

In our data we encountered 30-second intervals in which zero trade was recorded. This would make the formula (1) indeterminate for those ticks. To overcome this issue, the recent most value of $S_{VWAP}$ for each stock $k$ was considered for these ticks.

We include only 50 stocks in our analysis that had either no gap interval or very few gap intervals. In other words, these stocks are highly traded in the market.

Also, 2014 was the year when General Elections were held in India and a change in government was seen after 10 years. One may expect high volatility during the election times. We wanted to study how does our model gets impacted during the election or pre-election or the post-election periods. Thus our discrete time series data was studied under three periods: (a) pre-election period: Jan-Feb 2014 (b) election period: Mar-May 2014, (c) post-election period Jun-Dec 2014.

## 3) Methods and methodology

### 3.1 Pair copula construction

Before we explain the construction of the R-vine structure, it is important to have a clear understanding of a Copula. So we first recall some preliminaries. For any natural number $n$, let $I^n$ denote the unit cube in the extended $n$–dimensional space $\overline{\mathbb{R}}^n$. The elements of $\overline{\mathbb{R}}^n$ are $n$–tuples of extended real numbers $a_i$: $a = (a_1, \ldots, a_n)$. For any $a, b \in \overline{\mathbb{R}}^n$, we shall write $a \leq b$ whenever $a_i \leq b_i$ for all $i$. Now for any $a \leq b$, the Cartesian product of closed intervals, $B = [a_1, b_1] \times \ldots \times [a_n, b_n]$, is called an $n$–box and will be denoted by $[a,b]$. The set of vertices, $V$, of $B$ is the collection of all $n$–tuples $(c_1, \ldots, c_n)$ for which each $c_i = a_i$ or $b_i$. Let $H$ be a real valued function with domain of the form $S_1 \times \ldots \times S_n$, where each $S_i$ is a subset of extended real numbers $\overline{\mathbb{R}}$. The $H$–volume of $B$ is defined to be the sum

$$V_H(B) = \sum_{c \in V} sgn(c) H(c) \tag{3}$$

Here $sgn(c)$ takes on +1 if $c_i = a_i$ even number of times; and it takes on -1 otherwise. Also, note that the above summation is finite since the total number of vertices is finite. The reader is referred to [27] for other equivalent forms of $V_H(B)$.

An $n$–dimensional Copula is a function $C: I^n \rightarrow I$ satisfying the following axioms:

(i) $C\langle u \rangle = 0$ if there exists an $i$ such that $u_i = 0$.

(ii) $C\langle u \rangle = u_k$ if $u_i = 1$ for all $i \neq k$.

(iii) $V_C([a,b]) \geq 0$ for any $n$–box $[a,b]$ with $a,b \in I^n$.

A real valued function $F$ defined on $\overline{\overline{\mathbb{R}}}^n$ is called an $n$–dimensional distribution function if

(i) $V_F(B) \geq 0$ for all $n$–boxes $B$ with vertices in $\overline{\overline{\mathbb{R}}}^n$; and

(ii) $F\langle u \rangle = 0$ whenever $u_i = -\infty$ for some $i$.

(iii) $F(u) = 1$ whenever $u_i = \infty$ for all $i$.

It has been established in [27] that the $n$–dimensional distribution function $F$ has one dimensional marginal distribution functions $F_1, F_2, \ldots, F_n$.

The famous Sklar's theorem guarantees that there exists an $n$–dimensional Copula function $C$ such that $F(x_1, x_2, \ldots, x_n) = C(F_1(x_1), F_2(x_2), \ldots, F_n(x_n))$. However, we are more interested in the converse which states that for a given $n$–dimensional Copula function $C$ and univariate distribution functions $F_1, F_2, \ldots, F_n$, the formula $F(x_1, x_2, \ldots, x_n) = C(F_1(x_1), F_2(x_2), \ldots, F_n(x_n))$ defines an $n$–dimensional distribution function with marginals are $F_1, F_2, \ldots, F_n$. Equivalently the joint density function $f(x_1, x_2, \ldots, x_n) = f_1(x_1) f_2(x_2) \ldots f_n(x_n) c(F_1(x_1), F_2(x_2), \ldots, F_n(x_n))$ where $c$ is the $n$th order partial derivative of $C$. Thus, if we wish to study the joint behaviour of $n$ random variables, we can first fit the marginal distribution functions of each random variable separately and then combine them through an appropriate multivariate copula.

The process of constructing multivariate copula that we adopt is the Pair-wise Copula Construction (PCC) which relies on Vine copulas (or pair copulas) introduced in [7]. At the heart of this process lies the fact that a joint copula function is broken down as product of bivariate copula functions that can be estimated independently. Thus, bivariate copulas are building blocks for the PCC method.

An R-vine on $n$ variables as introduced by Bedford and Cooke [9] is a finite sequence of trees $T_j = (V_j, E_j)$, $j = 1, 2, \ldots, n-1$, with vertices $V_j$ and edges $E_j$ satisfying the conditions:

(i) The tree $T_1$ has nodes $N_1 = \{1, 2, \ldots, n\}$.

(ii) Trees $T_j$ are connected with nodes $N_j = E_{j-1}$ and that the cardinality of $N_j$ is $n-j+1$ for each $j = 1, 2, \ldots, n$.

(iii) Let $a = \{a_1, a_2\}$ and $b = \{b_1, b_2\}$ be two elements of $N_j$ ($2 \leq j \leq n-1$), then $\{a,b\} \in E_j$ provided that the cardinality of $a \cap b$ is exactly one.

The last axiom says that we will join two nodes by an edge only when these nodes interpreted as edges of the preceeding tree have exactly one node of the preceeding tree in common.

Bedford and Cooke [9] follow a convenient way of enumerating the nodes of trees in an R-vine structure in terms of conditioned and conditioning sets. For further details and illustrative examples the reader may refer to [9, 15].

We make use of the same enumeration strategy to write down the probability density function corresponding to the distribution realized by the R-vine copula structure for the portfolio of stocks.

In order to construct an R-vine structure of stocks, we start with a tree $T_1$ with $n$ nodes ($N_1$) represented by each stock and $E_1$ edges. In our analysis, we considered $T_1$ as minimum spanning tree network of stocks based on both mutual information metric (Eq 10) and Kendall's Tau based metric (Eq 11). Edge in $E_1$ is represented by a bivariate copula $C_{\{s(e),t(e)\}}$ where s(e),

and t(e), are nodes connected by the edge $e$. Then we move on to next tree, $T_2$ with the nodes set $N_2$ same as the edge set $E_1$. Each node in $T_2$ is thus represented by $C_{\{s(e),t(e)\}}$ and edge in $E_2$ is represented by conditional copula $C_{\{s(e),t(e)/D(e)\}}$ where $D(e)$ is the common node. Similarly we keep on building the trees $T_3, T_4, \ldots, T_n$.

Once we have constructed a R-vine structure on $n$ stocks with random variables $X_1, X_2, \ldots, X_n$, their joint density function with marginal density functions $f_1, f_2, \ldots, f_n$ is given by

$$f(x_1, x_2, \ldots, x_n) = \prod_{j=1}^{n} f_j(x_j) \prod_{i=1}^{n-1} \prod_{e \in E_i} c_{s(e),t(e)/D(e)}(F_{s(e)/D(e)}(x_{s(e)}), F_{t(e)/D(e)}(x_{t(e)})) \quad (4)$$

where $F_{s(e)/D(e)}$ is distribution function of conditional random variable $X_{s(e)/D(e)}$ and $C_{s(e),t(e)/D(e)}$ is second order partial derivative of copula connecting $X_{s(e)/D(e)}$ $and$ $X_{t(e)/D(e)}$. For example consider the joint density function of three random variables $f$ can be decomposed as $f_1 f_{2/1} f_{3/12}$ where $f_{2/.}$ denotes conditional density functions. We can further decompose conditional density function $f_{2/1}$ as

$$f_{2/1} = \frac{f_{12}}{f_1} = \frac{f_1 f_2 c_{12}}{f_1} = f_2 c_{12}, \quad (5)$$

where $f_{12}$ is joint density of variable 1 and 2, $c_{12}$ is the 2$^{nd}$ order derivative of copula $C_{12}$ connecting variable 1 and 2. Similarly, we have

$$f_{3/12} = \frac{f_{123}}{f_{12}} = \frac{f_{23/1} f_1}{f_{2/1} f_1} = \frac{f_{23/1}}{f_{2/1}} = \frac{f_{2/1} f_{3/1} c_{23/1}}{f_{2/1}} = f_{3/1} c_{23/1} \quad (6)$$

Thus, using Eqs (5) and (6) we have joint density function of 3 variables can be decomposed as $f_1 f_2 c_{12} f_{3/1} c_{23/1} = f_1 f_2 f_3 c_{12} c_{13} c_{23/1}$. The analogous R-vine copula is given in Fig 1.

For fast execution of statistical methods such as the Maximum likelihood estimate, Morales and Napoles et al. [28] proposed an efficient scheme of storing an R-vine on $n$–variables as an $n \times n$ lower triangular matrix $M = (m_{ij})$. The matrix $M$ has interesting properties such as each column has distinct elements; and deleting the first row and first column of $M$ yields a $(n-1)$–dimensional R-vine matrix.

The decomposition in Eq (4) now can be expressed in terms of the $R$-vine matrix:

$$f(x_1, x_2, \ldots, x_n) = \prod_{j=1}^{n} f_j(x_j) \prod_{k=n-1}^{1} \prod_{i=n}^{k+1} C_{m_{kk},m_{ik}|m_{i+1,k},\ldots,m_{n,k}}(F_{m_{kk}|m_{i+1,k},\ldots,m_{n,k}}, F_{m_{ik}|m_{i+1,k},\ldots,m_{n,k}}). \quad (7)$$

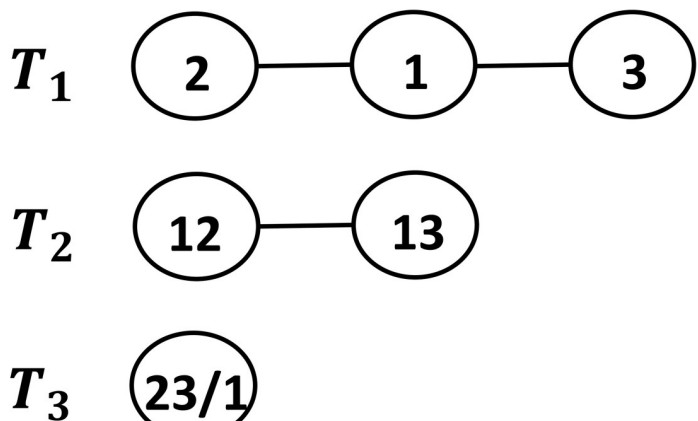

**Fig 1. R-vine copula with 3 stocks.** $T_1$, $T_2$, $T_3$ corresponds to trees 1, 2 and 3 respectively.

Note that the above equation is in terms of a bivariate copula function. An efficient algorithm for computing the conditional distributions appearing as arguments of this copula function has been proposed in [15].

## 3.2 Mutual information and Kendall's Tau based metrics

Mutual Information (MI) between two random variables captures mutual dependence between them and is zero if and only if they are independent. MI between two random variables is defined to be the difference between the sum of the respective entropies of random variables and their joint entropy.

The mutual information of discrete random variables $X$ and $Y$ is defined as

$$I(X, Y) = H(X) + H(Y) - H(X, Y) = \sum_i \sum_j f_{X,Y}(x_i, y_i) log\left(\frac{f_{X,Y}(x_i, y_i)}{f_X(x_i) f_Y(y_j)}\right) \tag{8}$$

A generalization to the continuous case is

$$I(X, Y) = \int \int f_{X,Y}(x, y) log\left(\frac{f_{X,Y}(x, y)}{f_X(x) f_Y(y)}\right) dx dy \tag{9}$$

Based on mutual information, the normalized distance [23] between two random variables $X$ and $Y$ is defined as

$$d(X, Y) = 1 - \frac{I(X, Y)}{H(X, Y)} \tag{10}$$

where, $I$ is the mutual information and $H$ is the joint entropy. Based on this metric, we can construct minimum spanning tree (MST) network between $n$ stocks. There are two well-known methods to construct Minimal Spanning Tree: Kruskal's algorithm and Prim's algorithm. We used Prim's algorithm for construction of the stock networks since the stocks networks are dense networks and in such cases Prim's algorithm works well.

We also considered building stock networks based on Kendall's Tau quantifier. The metric used is

$$d(X, Y) = (1 - |\tau_{X,Y}|), \tag{11}$$

where $\tau_{X,Y}$ is Kendall's Tau coefficient between $X$ and $Y$. Sharma and Habib [23] studied MI based stock networks and showed the existence nonlinearity in the stock returns data at high frequency level.

## 3.3 Fitting univariate models to log returns of stocks

A stochastic process $R_1, R_2, \ldots, R_t$ is a white noise process with mean μ and variance $\sigma^2$, if $E(R_t) = \mu$ for all $t$, $Var(R_t) = \sigma^2$ for all $t$, and $Cov(R_t, R_s) = 0$ for all $t \neq s$. In order to check if the log returns of stocks exhibit the properties of white noise, we carried out Ljung-Box test [29] to check if the log returns of stocks exhibit any autocorrelation or heteroscedasticity at 1% level of significance

$$H_0: \textit{no autocorrelation/heteroscedasticity}$$

$$H_A: \textit{autocorrelation/heteroscedasticity present}$$

We carried out hypothesis testing for each day and each stock on log returns and squares of log returns. Fig 2A corresponds to log returns and Fig 2B corresponds squares of log returns.

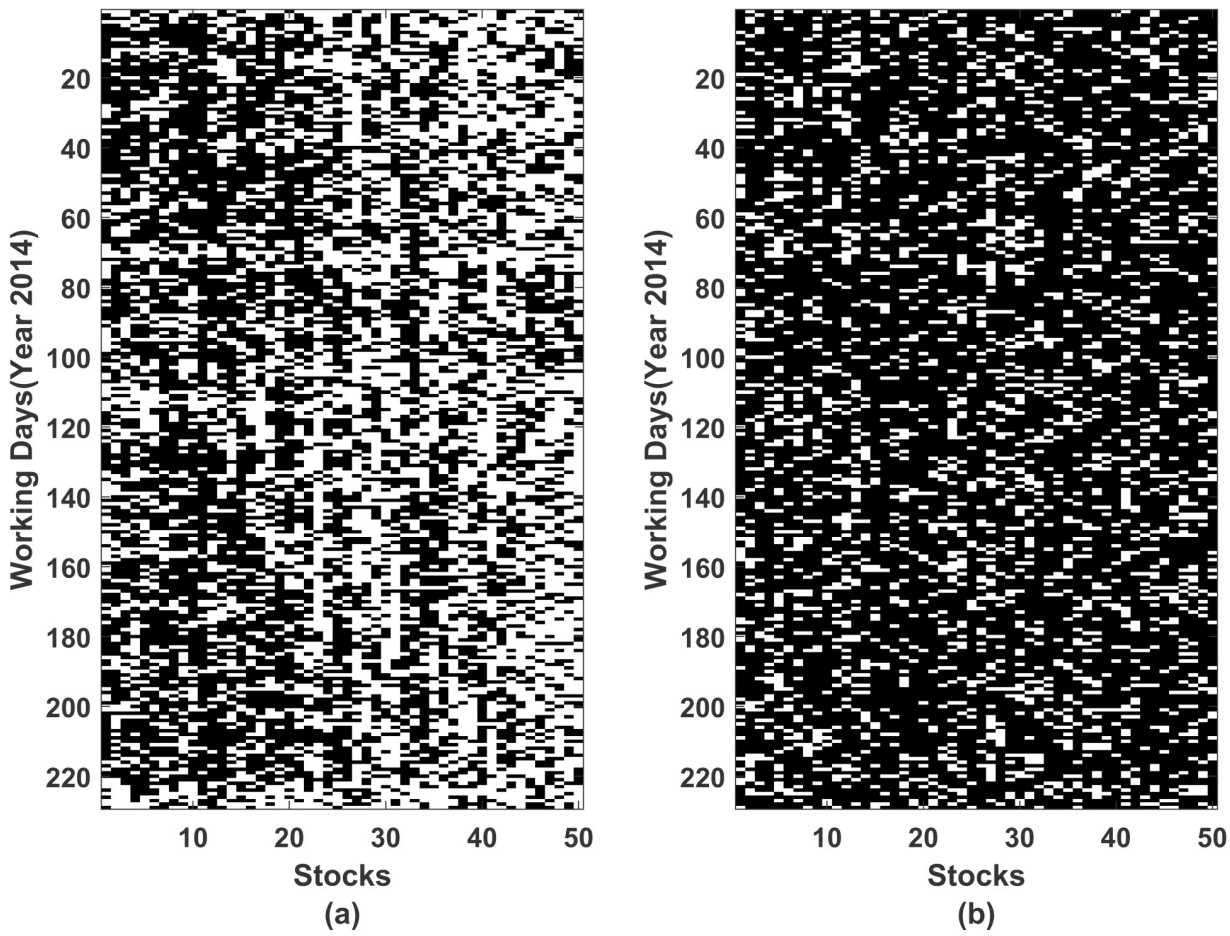

**Fig 2. Ljung-Box test on log returns and squares of log returns.** On horizontal *axis* we have listed 50 stocks and on vertical *axis* we have working days of year 2014. Black and white colour represents that the null hypothesis ($H_0$: *data is independent*, $H_A$: *data exhibit serial correlation*) is rejected or accepted respectively. (a) Corresponds to test applied to log returns (b) Corresponds to test applied to squares of log returns.

Presence of autocorrelation and heteroscedasticity can be seen at a lag of 1. Thus, GARCH methods are applied to our data aiming to remove the autocorrelation and heteroscedasticity in the time series. We tested for GARCH(1,1), ARMA(1,1)-GARCH(1,1) and ARMA(1,1)-EGARCH(1,1) models with the error estimated by student's t-distribution. A process $R_t$ is called an ARCH(p) process if $R_t = \mu + \sigma_t \varepsilon_t$ where $\varepsilon_t$ is a white noise and $\sigma_t = \sqrt{\omega + \sum_{i=1}^{p} \alpha_i R_{t-i}^2}$ is the conditional standard deviation of $R_t$ given the past values $R_{t-1}, \ldots, R_{t-p}$. It is to be noted that an ARCH(p) process has constant mean and constant unconditional variance but its conditional variance is not constant. The GARCH(p,q) model, on the other hand, tries to improve some of the deficiencies of the ARCH(p) model by expressing $\sigma_t$ in terms of the past values of standard deviation $\sigma_{t-1}, \ldots, \sigma_{t-q}$ in addition to the past values $R_{t-1}, \ldots, R_{t-p}$. Specifically, we have $\sigma_t = \sqrt{\omega + \sum_{i=1}^{p} \alpha_i R_{t-i}^2 + \sum_{j=1}^{q} \beta_i \sigma_{t-j}^2}$. For further details, the reader can refer to [30, 31].

We also tried fitting Normal Inverse Gaussian (NIG) distribution as well on the error terms. We used Kolmogorov Smirnov test to check the goodness of fit of univariate distribution on errors. Both NIG and student's t distribution turns out to be better choices over

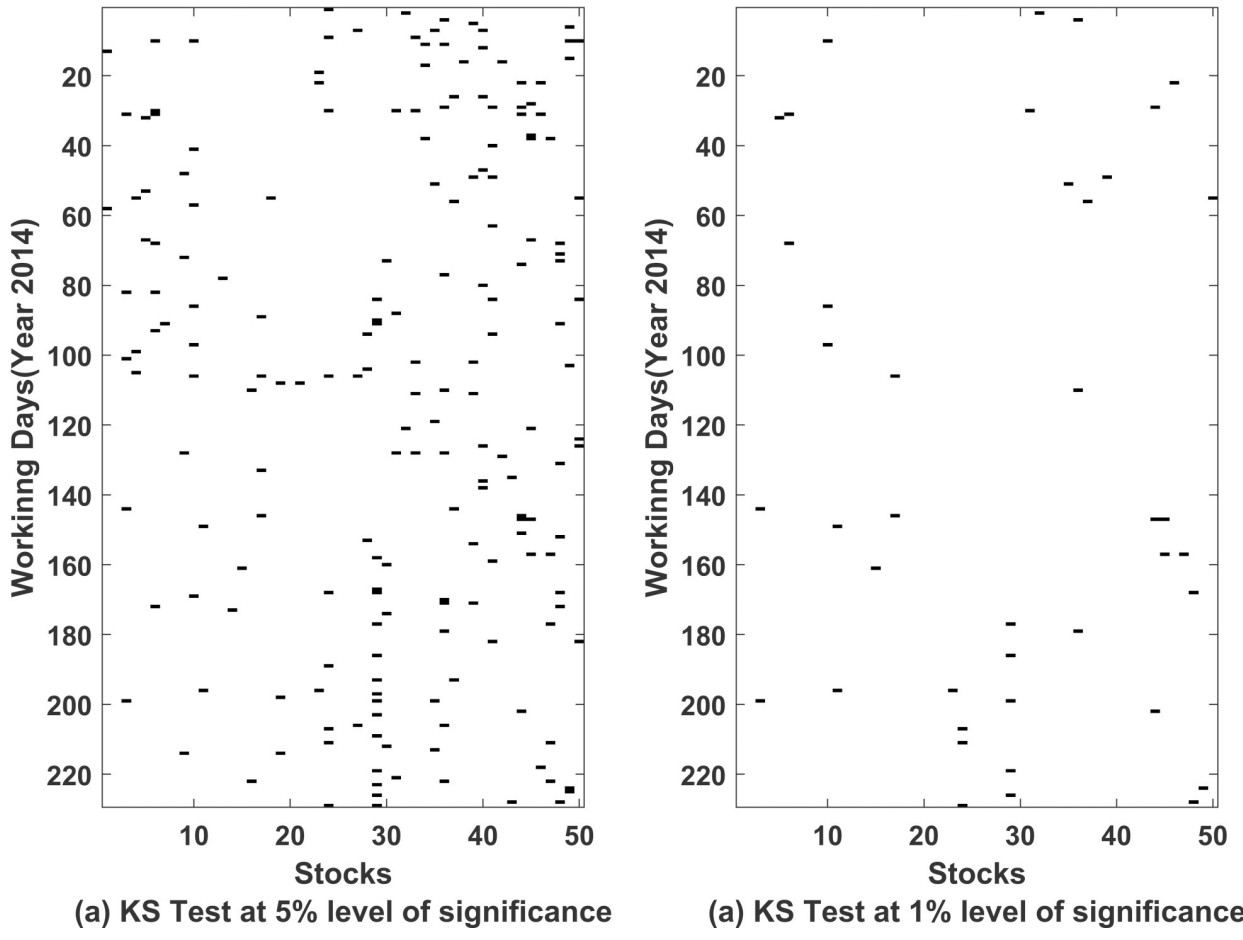

**Fig 3. Kolmogorov Smirnov test for testing t-distribution for error terms from ARMA(1,1)-EGARCH(1,1).** On horizontal *axis* we have listed 50 stocks and on vertical *axis* we have working days of year 2014. Black and white colour represents that the null hypothesis ($H_0$: *data follows t—distribution*) is rejected or accepted respectively. (a) Corresponds to test at 5% level of significance (b) Corresponds to test at 1% level of significance.

normal distribution. Due to computational simplicity, we used student's t distribution in our model. In Fig 3, we summarize the p-values corresponding to the test applied to the error terms obtained after fitting ARMA(1,1)-EGARCH(1,1) model for each of 50 stocks computed daily.

In all the equations given below, $R_{t,k}$ is as defined in Eq (2). The GARCH(1,1) model [31] for the $k$th stock is given by

$$R_{t,k} = \mu_k + \sigma_{t.k}\varepsilon_{t,k} \tag{12}$$

$$\sigma_{t,k}^2 = \omega_k + \beta_k \sigma_{t-1,k}^2 + \alpha_k \sigma_{t-1,k}^2 \varepsilon_{t-1,k}^2 \tag{13}$$

where, we fit a student's t-distribution to the noise $\epsilon_{t,k.}$

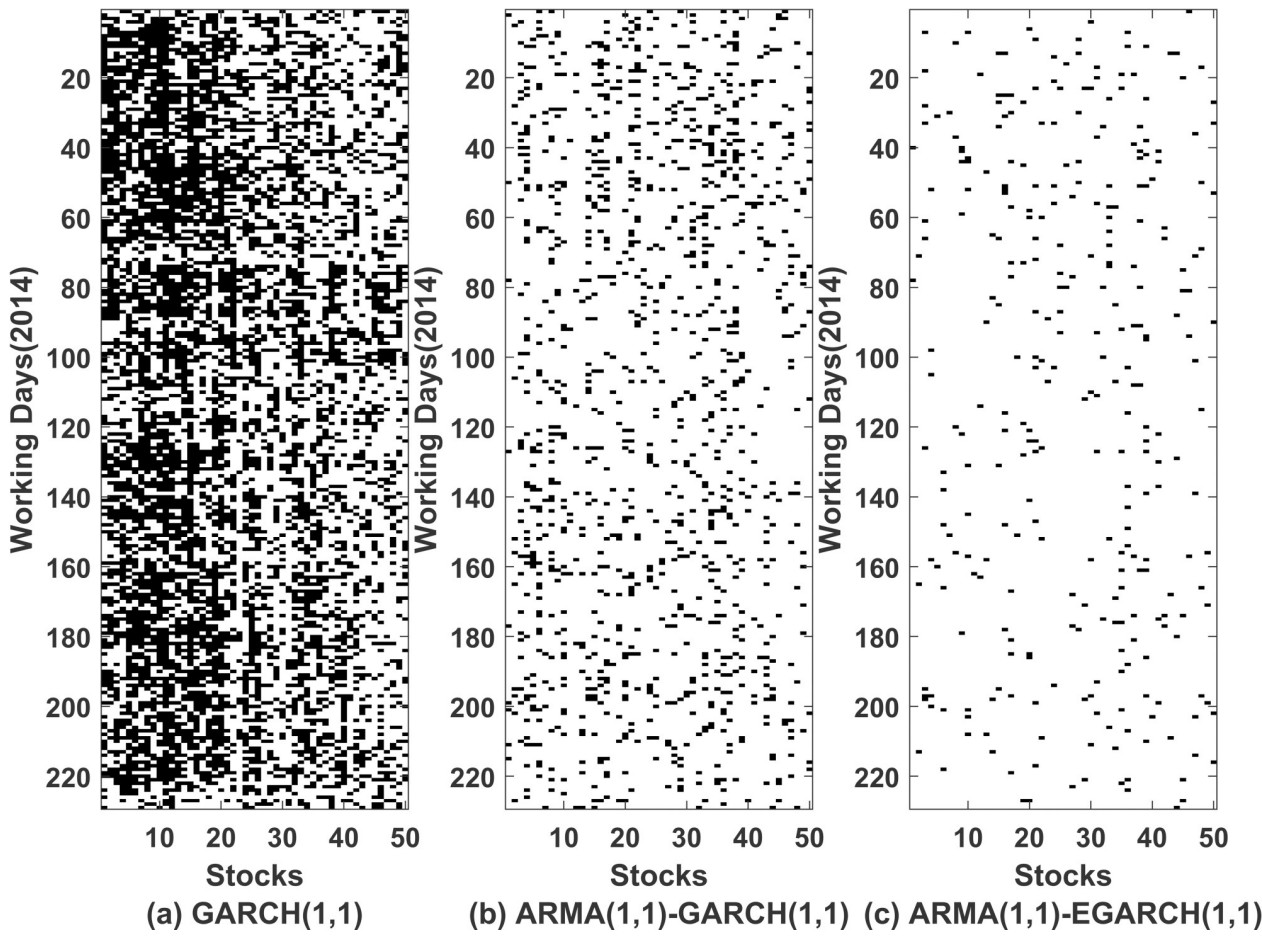

**Fig 4. Ljung-Box test on errors.** On horizontal *axis* we have listed 50 stocks and on vertical *axis* we have working days of year 2014. Black and white colour represents that the null hypothesis ($H_0$: *data is independent*, $H_A$: *data exhibit serial correlation*) is rejected or accepted respectively. (a), (b) and (c) corresponds to test applied to the error terms($\epsilon_{t,k}$) in GARCH(1,1), ARMA(1,1)-GARCH(1,1) and ARMA(1,1)-EGARCH(1,1) models respectively.

The ARMA(1,1)-GARCH(1,1) model [31] for the *k*th stock is given by

$$R_{t,k} = \mu_k + ar1(R_{t-1,k} - \mu_k) + ma1\sigma_{t-1,k}\varepsilon_{t-1,k} + \sigma_{t,k}\varepsilon_{t,k} \tag{14}$$

$$\sigma_{t,k}^2 = \omega_k + \beta_k\sigma_{t-1,k}^2 + \alpha_k\sigma_{t-1,k}^2\varepsilon_{t-1,k}^2 \tag{15}$$

where, we fit a student's t-distribution to the noise $\epsilon_{t,k}$.

The ARMA(1,1)-EGARCH(1,1) model [31] for the *k*th stock is given by

$$R_{t,k} = \mu_k + ar1(R_{t-1,k} - \mu_k) + ma1\sigma_{t-1,k}\varepsilon_{t-1,k} + \sigma_{t,k}\varepsilon_{t,k} \tag{16}$$

$$ln(\sigma_{t,k}^2)\omega_k + \gamma_k|\varepsilon_{t-1,k}| - \gamma_k[E(|\varepsilon_{t-1,k}|)] + \alpha_k\varepsilon_{t-1,k} + \beta_k ln(\sigma_{t-1}^2, k) \tag{17}$$

where we fit a student's t-distribution to the noise $\epsilon_{t,k}$.

In all three models, we tested if the noise term $\epsilon_{t,k}$ exhibit properties of a white noise by again running Ljung Box Tests at 1% level of significance. Figs 4 and 5 corresponds to the results obtained from running Ljung Box Test on $\epsilon_{t,k}$ and $\epsilon_{t,k}^2$ respectively. Clearly ARMA (1,1)-EGARCH(1,1) proves to be better fitted model in comparison to other models. We use

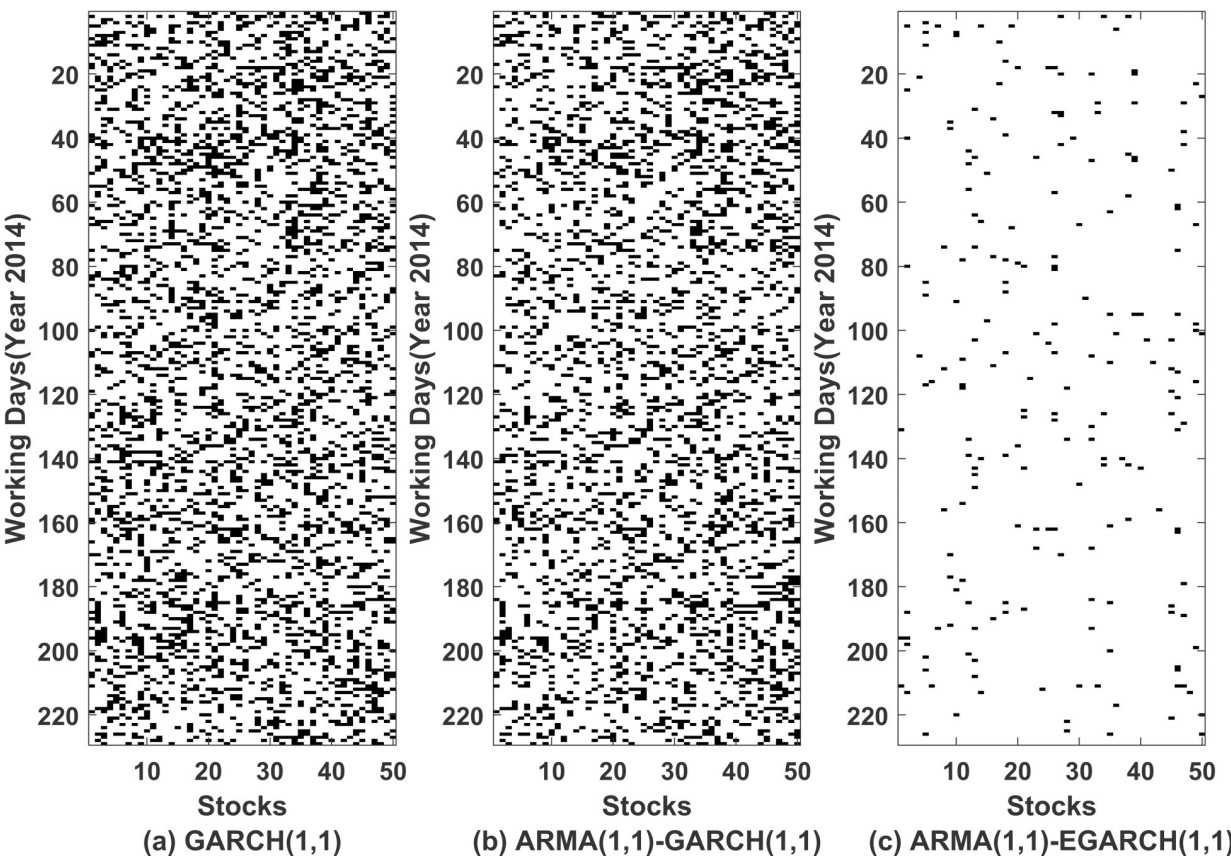

**Fig 5. Ljung-Box test on squares of errors.** On horizontal *axis* we have listed 50 stocks and on vertical *axis* we have working days of year 2014. Black and white colour represents that the null hypothesis ($H_0$: *data is independent*,$H_A$: *data exhibit serial correlation*) is rejected or accepted respectively. (a), (b) and (c) corresponds to test applied to the squares of error terms($\epsilon_{t,k}^2$) in GARCH(1,1), ARMA(1,1)-GARCH(1,1) and ARMA(1,1)-EGARCH(1,1) models respectively.

AIC values to compare the three methods. 94.84% of the times ARMA(1,1)-EGARCH(1,1) was seen to have the lowest AIC values and it again emerged to be a better fit in comparison to the other two methods. We used adjusted Pearson chi-squared goodness of fit test [32] to check the effectiveness of the univariate model for each stock on each working day at 5% and 1% level of significance. Fig 6 gives whether the null hypothesis, $H_0$: $ARMA(1,1) - EGARCH$ (1,1) *is a good fit*, was rejected (black colour) or accepted (white colour) for each stock on for each working day. 32 stocks out of 50 were seen to pass the test for more than 90% of times, i.e. null hypothesis was not rejected at 1% level of significance more than 90% of times. Also all the stocks showed an efficiency of a good fit for more than 72% of times. Thus, we conclude that ARMA(1,1)-EGARCH(1,1) is a good fit.

### 3.4 Value at risk (VaR) prediction

Value at risk (VaR) of a portfolio is measure of risk associated with it. For example if a portfolio has one-tick 5% VaR of $x$ amount, then it means that there is 5% chance that the portfolio looses its value by an amount $x$ over the time duration of one tick in the absence of trading. It is well known that $\alpha$% VaR of the portfolio is given by the $\alpha$–percentile of log returns of the portfolio [25]. Once the joint distribution function of $n$ stocks is known, we can use a Monte-Carlo simulation to estimate the VaR of the underlying portfolio. In this paper we have drawn inferences by calculating the VaR for equally weighted portfolios of 5, 10, 25, and 50 stocks.

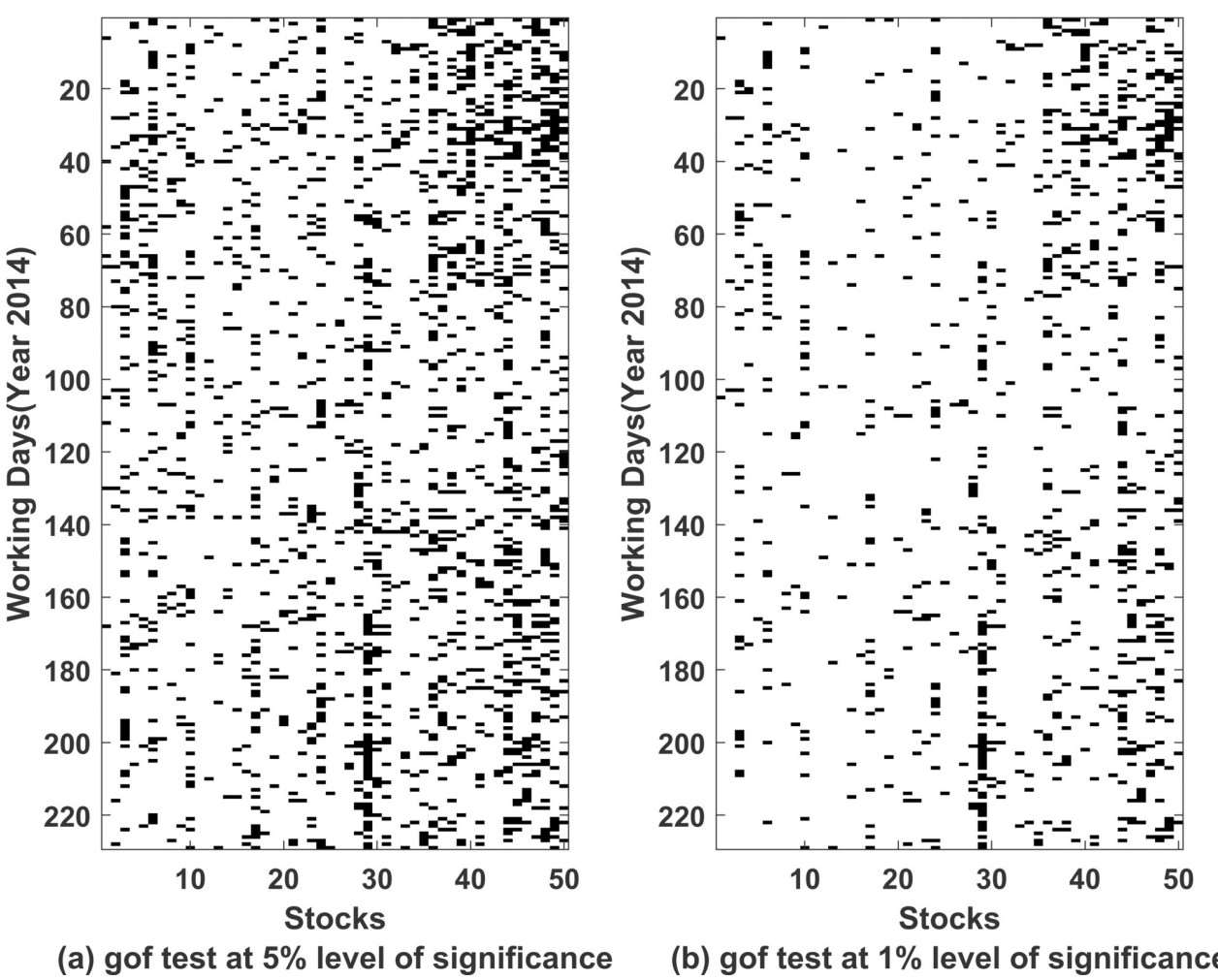

**Fig 6. Goodness of fit test for ARMA(1,1)-EGARCH(1,1) model.** On horizontal *axis* we have listed 50 stocks and on vertical *axis* we have working days of year 2014. Black and white colour represents that the null hypothesis ($H_0$: $ARMA(1,1)-EGARCH(1,1)$ is a good fit) is rejected or accepted respectively. (a) Corresponds to test at 5% level of significance (b) Corresponds to test at 1% level of significance.

Consider a portfolio consisting of $n$ stocks and random variables $S_{VWAP}(t,k)$, $R_{t+1,k}$ are as defined in Eqs (1) and (2). Let $w_k$ be the weight associated with $k$th stock in the portfolio and $S_{t,P}$ be the value of the portfolio corresponding to the tick $t$, then, the log return of the portfolio $R_{t+1,P}$ in the time interval $[t, t + 1]$ is given by

$$R_{t+1,P} = ln(\sum_{k=1}^{n} w_k e^{R_{t+1,k}}) \tag{18}$$

Using identities $e^x \sim (1 + x)$, and $\ln(1 + x) \sim x$ for small $x$, in above equation, we get

$$R_{t+1,P} \cong \sum_{i=1}^{n} w_i R_{t+1,i} \tag{19}$$

We first use ARMA(1,1)+EGARCH(1,1) to model univariate log returns of each stock and then use R-Vine copula construction on the error terms $\epsilon_{t,k}$ to estimate joint copula on the error terms. We fit the model on the first 4 hours of each day and use it to predict Var for next 2 hours. We summarize the algorithm as below:

1. Consider log returns of each stock for the first 4 hours i.e. 9:30AM to 1:30PM (this gives 480 terms in each time series) on each day.

2. Fit an ARMA(1,1)-EGARCH(1,1) model to log returns of each stock obtained in step 1, with univariate Student's t-distribution assumed on the error term $\in_{t,k}$ of each stock $k$. So if there were $n$–stocks in the portfolio then the data generated at this step can be written conveniently as $(\in_{t,k})_{480 \times n}$.

3. Fit an R-vine copula structure to the random variables $\epsilon_{t,1}, \epsilon_{t,2}, \ldots, \epsilon_{t,n}$ (sampled at 480 ticks in step 2) to obtain the joint distribution of the error terms. In the R-vine algorithm we choose the first tree $T_1$ as the minimum spanning tree based on Kendall's Tau metric (Eq 11) and also MI based metric (Eq 10). In this paper we fitted the R-vine structure on n = 50 stocks.

4. Using the joint distribution obtained in step 3, we then employ Monte-Carlo simulation to generate a large number of values (say N = 5000) of $(\epsilon_{481,1}, \epsilon_{481,2}, \ldots, \epsilon_{481,n})$ simultaneously and substitute these in Eqs 16 and 17 to estimate the corresponding large number of values of $(R_{481,1}, R_{481,2}, \ldots, R_{481,n})$. For each of the N tuples $(R_{481,1}, R_{481,2}, \ldots, R_{481,n})$ obtained, compute the portfolio log return $R_{481,P}$ using Eq 19. In our analysis, we have worked with equally weighted portfolios with 5,10,25,50 stocks respectively.

5. The $\alpha$% VaR for $481st$ instant, $VaR_{481,P}$ is now calculated by finding the $\alpha$ percentile of the N simulated values of $R_{481,P}$. Here P a portfolio whose size is chosen to be of 5, 10, 25, and 50 stocks respectively. In this paper we have considered $\alpha$ = 5%, 10% respectively.

6. We then compare the actual $R_{481,P}$ with the estimated $VaR_{481,P}$.

7. Once the actual $R_{481,k}$ is known, then we can use Eq (16) to calculate actual $\epsilon_{481,P}$ as

$$actual\ \varepsilon_{481,k} = \frac{R_{481,k} - (\mu_k + ar1(R_{480,k} - \mu_k) + ma1\sigma_{480,k}\varepsilon_{480,k})}{\sigma_{481,k}}.$$

Next, we use Eq (17) to calculate $\sigma_{482,k}$. We then repeat steps 4, 5 and 6 for predicting $(R_{482,1}, R_{482,2}, \ldots, R_{482,n})$ and compare the actual $R_{482,P}$ with the estimated $VaR_{482,P}$. This way we calculate estimated $VaR_{i,p}$ where $i$ = 483,...,719, and compare these values with the respective actual $R_{i,p}$. Note that the model was fitted only once a day.

8. We repeat step 1 to step 7 for all working days in year 2014.

## 4) Discussion

Data for each day was divided into 2 subsets: training data from 9:30AM to 1:30PM and testing data 1:31PM to 3:30PM. We fit both Kendall Tau's based and MI based vine copula structure on the training data as discussed in the previous section. We then estimated VaRs corresponding to equally weighted portfolios for each time tick of the testing data. In our analysis we have considered portfolios consisting of all 50 stocks, randomly picked 25 or 10 or 5 stocks. Also, we have considered 5% and 10% VaRs in all the cases. To check the effectiveness of our model we carried out unconditional (UC) and conditional (CC) coverage test formulated by Christoffersen [33].

$$H_o \text{ for UC Test: Correct exceedance}$$

$$H_o \text{ for CC Test: Correct exceedance and Independence.}$$

**Table 1. Unconditional (UC) and conditional (CC) coverage tests at 1% and 5% level of significance: Pre-election period.**

| Pre-election Period | 5 stocks 10% VaR | 10 stocks 10% VaR | 25 stocks 10% VaR | 50 stocks 10% VaR | 5 stocks 5% VaR | 10 stocks 5% VaR | 25 stocks 5% VaR | 50 stocks 5% VaR |
|---|---|---|---|---|---|---|---|---|
| UCpvalue > 0.01 (Kendall's Tau method) | 87.50% | 84.38% | 78.13% | 87.50% | 87.50% | 90.63% | 84.38% | 84.38% |
| UCpvalue > 0.01 (MI method) | 87.50% | 87.50% | 81.25% | 87.50% | 87.50% | 90.63% | 84.38% | 87.50% |
| UCpvalue > 0.05 (Kendall's Tau method) | 75.00% | 84.38% | 71.88% | 75.00% | 81.25% | 84.38% | 75.00% | 78.13% |
| UCpvalue > 0.05 (MI method) | 75.00% | 84.38% | 71.88% | 75.00% | 81.25% | 84.38% | 78.13% | 78.13% |
| CCpvalue > 0.01 (Kendall's Tau method) | 90.63% | 78.13% | 62.50% | 43.75% | 87.50% | 81.25% | 78.13% | 68.75% |
| CCpvalue > 0.01 (MI method) | 90.63% | 78.13% | 65.63% | 43.75% | 87.50% | 81.25% | 78.13% | 71.88% |
| CCpvalue > 0.05 (Kendall's Tau method) | 71.88% | 71.88% | 37.50% | 28.13% | 84.38% | 78.13% | 65.63% | 62.50% |
| CCpvalue > 0.05 (MI method) | 71.88% | 71.88% | 43.75% | 21.88% | 84.38% | 78.13% | 65.63% | 59.38% |

There are 32, 39 and 113 days in the pre-election, election and post-election period for which our proposed model was a good fit. We carried out the hypothesis testing for each day and calculated percentage of times, the null hypothesis was not rejected. We refer to this calculated percentage of times as the success rate of the model. Tables 1–3 summarizes the results obtained in pre-election, election and post-election period.

It was observed that the VaR prediction were more accurate in case of portfolios consisting of small number of stocks like 5 or 10 in comparison to portfolios consisting of large number of stocks like 25 or 50. Also, the success rate of MI based model was seen to be much better than the Kendall's Tau based model, 41 out of 96 times (42.71%) in comparison to 6 out of 96 (6.25%) times when success rate of Kendall's Tau based model was observed to be better than that of MI based model. 49 out of 96 times (51.04%), the success rates based on both the methods were seen to be at par. One can also observe that even during the election times which is full of uncertainties, the success rate of the model was quite high.

## 5) Conclusion

This paper demonstrates the power of incorporating mutual information based metrics into the construction of R-vine copula structures in learning the joint distribution of a large

**Table 2. Unconditional (UC) and conditional (CC) coverage tests at 1% and 5% level of significance: Election period.**

| Election Period | 5 stocks 10% VaR | 10 stocks 10% VaR | 25 stocks 10% VaR | 50 stocks 10% VaR | 5 stocks 5% VaR | 10 stocks 5% VaR | 25 stocks 5% VaR | 50 stocks 5% VaR |
|---|---|---|---|---|---|---|---|---|
| UCpvalue > 0.01 (Kendall's Tau method) | 92.31% | 89.74% | 89.74% | 87.18% | 92.31% | 92.31% | 87.18% | 87.18% |
| UCpvalue > 0.01 (MI method) | 92.31% | 87.18% | 89.74% | 87.18% | 92.31% | 92.31% | 92.31% | 89.74% |
| UCpvalue > 0.05 (Kendall's Tau method) | 71.79% | 82.05% | 82.05% | 79.49% | 87.18% | 89.74% | 87.18% | 76.92% |
| UCpvalue > 0.05 (MI method) | 79.49% | 82.05% | 82.05% | 79.49% | 87.18% | 92.31% | 87.18% | 79.49% |
| CCpvalue > 0.01 (Kendall's Tau method) | 89.74% | 82.05% | 82.05% | 56.41% | 92.31% | 87.18% | 87.18% | 69.23% |
| CCpvalue > 0.01 (MI method) | 89.74% | 84.62% | 82.05% | 58.97% | 92.31% | 89.74% | 89.74% | 74.36% |
| CCpvalue > 0.05 (Kendall's Tau method) | 64.10% | 69.23% | 56.41% | 38.46% | 82.05% | 84.62% | 69.23% | 48.72% |
| CCpvalue > 0.05 (MI method) | 71.79% | 71.79% | 56.41% | 35.90% | 84.62% | 87.18% | 74.36% | 51.28% |

**Table 3. Unconditional (UC) and conditional (CC) coverage tests at 1% and 5% level of significance: Post-election period.**

| Election Period | 5 stocks 10% VaR | 10 stocks 10% VaR | 25 stocks 10% VaR | 50 stocks 10% VaR | 5 stocks 5% VaR | 10 stocks 5% VaR | 25 stocks 5% VaR | 50 stocks 5% VaR |
|---|---|---|---|---|---|---|---|---|
| UCpvalue > 0.01 (Kendall's Tau method) | 91.15% | 88.50% | 81.42% | 81.42% | 90.27% | 89.38% | 83.19% | 81.42% |
| UCpvalue > 0.01 (MI method) | 91.15% | 88.50% | 83.19% | 81.42% | 91.15% | 89.38% | 85.84% | 83.19% |
| UCpvalue > 0.05 (Kendall's Tau method) | 78.76% | 81.42% | 74.34% | 69.91% | 82.30% | 81.42% | 74.34% | 74.34% |
| UCpvalue > 0.05 (MI method) | 80.53% | 82.30% | 74.34% | 72.57% | 84.07% | 82.30% | 74.34% | 75.22% |
| CCpvalue > 0.01 (Kendall's Tau method) | 88.50% | 82.30% | 69.03% | 54.87% | 89.38% | 85.84% | 70.80% | 66.37% |
| CCpvalue > 0.01 (MI method) | 88.50% | 84.07% | 68.14% | 55.75% | 89.38% | 86.73% | 70.80% | 69.03% |
| CCpvalue > 0.05 (Kendall's Tau method) | 76.11% | 73.45% | 49.56% | 37.17% | 82.30% | 77.88% | 57.52% | 53.98% |
| CCpvalue > 0.05 (MI method) | 76.99% | 73.45% | 49.56% | 36.28% | 84.07% | 78.76% | 61.06% | 56.64% |

number of stocks from a high frequency market data. The data considered in the present analysis has an instant-by-instant record of transactions of 89 stocks listed on the National Stock Exchange (NSE) of India in the year 2014. In order to give a time series interpretation to our data, we divide each working day into 720 "ticks" where each tick represents a 30 second duration. Out of the 89 stocks, we have considered only the top 50 traded stocks. On the basis of the Ljung-Box test it is concluded that ARMA(1,1)-EGARCH(1,1) captured the autocorrelation and heteroscedasticity of the time series of log returns of the above portfolio of 50 stocks significantly better than the famous GARCH(1,1) and ARMA(1,1)-GARCH(1,1) methods. In fact on 94.84% of the occasions the AIC values obtained after fitting ARMA(1,1)-EGARCH (1,1) were found to be the lowest in comparison to the other methods (In the R software package, a lower AIC indicates that the model is superior). The joint distribution of the respective error terms in the ARMA(1,1)-EGARCH(1,1) model applied to each stock is then computed by learning R-Vine copula structures in 2 ways: first, by starting with the minimal spanning tree computed on the basis of the mutual information metric; and second, by starting with the minimal spanning tree computed on the basis of the Kendall's Tau based metric. Next, the VaR of the underlying 50 stock portfolio is computed through Monte-Carlo simulations in both the cases. The Christoffersen's UC and CC tests show that VaR predictions in the mutual information case out performs the VaR predictions in the Kendall's Tau case. The success rate obtained from the MI based method is seen to be higher than Kendall's Tau based method on 42.71% occasions. On 51.04% of the occasions the success rates from both the methods were at par. The predictions were quite good even during the election period when there is lot of anticipation amongst the buyers.

We finally conclude that MI based R-Vine Copula model is able to capture the joint distribution well and thus leads to better VaR predictions in a high frequency scenario.

## Supporting information

**S1 Table. Data corresponding to Figs 2–6 and Tables 1–3.**
(XLSX)

## Acknowledgments

We thank the Shiv Nadar University for providing the computational facilities and the necessary infrastructure needed to carry out the present research. Both authors extend a special

gratitude to Professor Amber Habib for his encouragement and valuable comments. The authors also thank the reviewers for their valuable suggestions, which have enhanced the clarity of the paper significantly.

## Author Contributions

**Conceptualization:** Charu Sharma, Niteesh Sahni.

**Data curation:** Charu Sharma.

**Formal analysis:** Charu Sharma, Niteesh Sahni.

**Investigation:** Charu Sharma, Niteesh Sahni.

**Methodology:** Charu Sharma, Niteesh Sahni.

**Project administration:** Charu Sharma, Niteesh Sahni.

**Resources:** Charu Sharma, Niteesh Sahni.

**Software:** Charu Sharma, Niteesh Sahni.

**Supervision:** Charu Sharma.

**Validation:** Charu Sharma.

**Visualization:** Charu Sharma.

**Writing – original draft:** Charu Sharma, Niteesh Sahni.

**Writing – review & editing:** Charu Sharma, Niteesh Sahni.

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
