## [Decision Letter · Decision Letter 0]

15 Apr 2021

PONE-D-20-23135

A Mutual Information Based R-Vine Copula Strategy To Estimate VaR in High Frequency Stock Market Data

PLOS ONE

Dear Dr. Sharma,

Thank you for submitting your manuscript to PLOS ONE. After careful consideration, we feel that it has merit but does not fully meet PLOS ONE’s publication criteria as it currently stands. Therefore, we invite you to submit a revised version of the manuscript that addresses the points raised during the review process.

We look forward to receiving your revised manuscript.

Kind regards,

Alessandro Barbiero, Ph.D. in Statistics

Academic Editor

PLOS ONE

Journal Requirements:

We note that you have indicated that data from this study are available upon request. PLOS only allows data to be available upon request if there are legal or ethical restrictions on sharing data publicly. For more information on unacceptable data access restrictions, please see http://journals.plos.org/plosone/s/data-availability#loc-unacceptable-data-access-restrictions.

2a) If there are ethical or legal restrictions on sharing a de-identified data set, please explain them in detail (e.g., data contain potentially sensitive information, data are owned by a third-party organization, etc.) and who has imposed them (e.g., an ethics committee). Please also provide contact information for a data access committee, ethics committee, or other institutional body to which data requests may be sent.

2b) If there are no restrictions, please upload the minimal anonymized data set necessary to replicate your study findings as either Supporting Information files or to a stable, public repository and provide us with the relevant URLs, DOIs, or accession numbers. For a list of acceptable repositories, please see http://journals.plos.org/plosone/s/data-availability#loc-recommended-repositories.

Additional Editor Comments:

- I would suggest a brief description of ARMA and GARCH models and the meaning of their parameters. How did you check the fit of these models, just using AIC?

- For the univariate distribution, you explained you considered both NIG and Student's t and you selected the latter due to its simplicity. What do you specifically mean? That it is easier to estimate its parameters? How is Kolmogorov-Smirnov test affected by the fact that parameters are unknown and need to be estimated? As far as I know, estimating unknown parameters may lead to a "biased" p-value. Did you explored other parametric families (eg stable distributions)?

- Figures are blurred, please fix them

- Perhaps when explaining the application of vines some graphs could be beneficial to a better understanding

Reviewers' comments:

Reviewer's Responses to Questions

**Comments to the Author**

1. Is the manuscript technically sound, and do the data support the conclusions?

Reviewer #1: Yes

2. Has the statistical analysis been performed appropriately and rigorously? 

Reviewer #1: Yes

3. Have the authors made all data underlying the findings in their manuscript fully available?

Reviewer #1: Yes

4. Is the manuscript presented in an intelligible fashion and written in standard English?

Reviewer #1: Yes

5. Review Comments to the Author

Reviewer #1: The paper seems quite polished. The authors clearly know this material, and then seem to have executed everything correctly. I would "Accept," but with one minor suggestion. The authors repeated refer to "mutual information," which is not a phrase with which I'm familiar. I believe they mean "dependence," in the statistical sense. Could they more clearly define (very early in the paper) what they mean by "mutual information"?

6. PLOS authors have the option to publish the peer review history of their article (what does this mean?). If published, this will include your full peer review and any attached files.

Reviewer #1: **Yes: **David Zimmer

---

## [Author Response · Author response to Decision Letter 0]

1 Jun 2021

Dear Editor,

Our sincere thanks to you and all the reviewers for your valuable comments on the scientific content and presentation of our manuscript titled “A Mutual Information Based R-Vine Copula Strategy To Estimate VaR in High Frequency Stock Market Data” (PONE-D-20-23135). We have made several changes to manuscript in accordance with these comments.

We hope that the revised version of the manuscript will be considered favourably by PLOS ONE. We now give an item wise response to the reviewers’ comments:

Regarding Journal Requirements:

We crosschecked and the revised manuscript is in accord with the requirements.

2) Restriction on sharing data

We have uploaded the processed data as a supporting file “S1 Table”. The raw data is “available from a third party”, details of which are provided in the cover letter as well: NSE Data & Analytics (https://www.nseindia.com/supra_global/content/dotex/about_dotex.htm). . The terms of purchase prohibit us from redistributing the Historical Data or any component of it.

Revised Data Availability Statement:

Raw data cannot be shared publicly because of copyright agreement with NSE Data & Analytics, formerly known DotEx International Ltd.(third party). Data are available from the NSE Data & Analytics, formerly known DotEx International Ltd., (contact via dotex_kraops@nse.co.in) for researchers who meet the criteria for access to confidential data. 

The terms of purchase prohibit us from redistributing the Historical Data or any component of it. However, the raw data can be purchased from NSE Data & Analytics at: https://www.nseindia.com/supra_global/content/dotex/data_products.htm

We also confirm that we did not have any special access privileges that others would not have. 

However, we also confirm that all the processed data corresponding to each figure and each table is uploaded as a supporting file named “S1 Table”.

While adhering to the editor’s suggestion of including a brief description of ARCH and GARCH, we needed to include a fresh reference of a paper by Robert F. Engle. The list of references have been revised in view of the stated changes and we are certain that no references correspond to retracted manuscripts. Further, we have updated the missing DOIs and ISBN(in case of books) information as well. All the relevant changes have been incorporated in the revised document.

Additional Editor Comments:

1) I would suggest a brief description of ARMA and GARCH models and the meaning of their parameters. How did you check the fit of these models, just using AIC?

Specific details about ARCH and GARCH models have been inserted in section 3.1(lines 271-279) of the manuscript.

Fitting of the three models i.e. GARCH(1,1), ARMA(1,1)-GARCH(1,1) and ARMA(1,1)-EGARCH(1,1) were based on the two steps. First step: p-values for each parameter in each model was observed to be significantly high justifying the fit of the respective models. Second step: to pick the best model out of the three we used minimum AIC criteria.

2) For the univariate distribution, you explained you considered both NIG and Student's t and you selected the latter due to its simplicity. What do you specifically mean? That it is easier to estimate its parameters? How is Kolmogorov-Smirnov test affected by the fact that parameters are unknown and need to be estimated? As far as I know, estimating unknown parameters may lead to a "biased" p-value. Did you explored other parametric families (eg stable distributions)?

NIG is a subclass of generalized hyperbolic distribution depending upon 4 parameters. On the other hand, Student’s t-distribution depends only on 1 parameter. Considering errors associated with the estimation of the parameters, it is always beneficial to estimate one parameter over 4 parameters. We tried fitting of both the distributions on the univariate data and ran Kolmogrov Smirnov test to check goodness of fit. All 50 stocks cleared the test at 1% level of significance in both the cases. To keep our computational errors low we decided to pick Student’s t-distribution over NIG.

Kolmogrov-Smirnov test was conducted using simulations, which ensured unbiased p-values.

Since NIG and t-distribution gave us satisfying results, thus we decided to stick to them and didn’t explore further.

3) Figures are blurred, please fix them

We have re-generated all figures at 600dpi and ran through the PACE software. Hope they are of acceptable quality now.

4) Perhaps when explaining the application of vines some graphs could be beneficial to a better understanding

Thank you for your valuable suggestion. We have now included an example in section 3.1(refer to lines 204-213 of the manuscript) along with the associated graph (Fig 1).

Reviewer #1:

1) …one minor suggestion. The authors repeated refer to "mutual information," which is not a phrase with which I'm familiar. I believe they mean "dependence," in the statistical sense. Could they more clearly define (very early in the paper) what they mean by "mutual information"?

Thanks for your valuable suggestion. We have now included the definition of mutual information in the introduction section 1 (lines 69-74) and explained it in detail in section 3.2(lines 232-244).

---

## [Editor Report · Decision Letter 1]

3 Jun 2021

A Mutual Information Based R-Vine Copula Strategy To Estimate VaR in High Frequency Stock Market Data

PONE-D-20-23135R1

Dear Dr. Sharma,

We’re pleased to inform you that your manuscript has been judged scientifically suitable for publication and will be formally accepted for publication once it meets all outstanding technical requirements.

Kind regards,

Alessandro Barbiero, Ph.D. in Statistics

Academic Editor

PLOS ONE

Additional Editor Comments (optional):

Dear authors,

I would just ask you to revise the new part explaining Mutual information (rows 69-72): it is still a bit confusing with some repetitions ("random variables")
---

## [Editor Report · Acceptance letter]

7 Jun 2021

PONE-D-20-23135R1 

A Mutual Information Based R-Vine Copula Strategy To Estimate VaR in High Frequency Stock Market Data 

Dear Dr. Sharma:

I'm pleased to inform you that your manuscript has been deemed suitable for publication in PLOS ONE. Congratulations! Your manuscript is now with our production department. 

Kind regards, 

on behalf of

Dr. Alessandro Barbiero 

Academic Editor

PLOS ONE